# An Efficient Hybrid Job Scheduling Optimization (EHJSO) approach to enhance resource search using Cuckoo and Grey Wolf Job Optimization for cloud environment

**D. Paulraj[1], T. Sethukarasi[1], S. Neelakandan[1], M. Prakash[2], E. Baburaj[3]***

**1** Department of Computer Science and Engineering, R.M.K. Engineering College, Chennai, India, **2** School of Computing Science and Engineering, VIT University, Chennai, India, **3** Department of Electrical and Computer Engineering, Bule Hora University, Bule Hora, Ethiopia

* baburajcse@bhu.edu.et

**Data Availability Statement:** All relevant data are within the paper.

## Abstract

Cloud computing has now evolved as an unavoidable technology in the fields of finance, education, internet business, and nearly all organisations. The cloud resources are practically accessible to cloud users over the internet to accomplish the desired task of the cloud users. The effectiveness and efficacy of cloud computing services depend on the tasks that the cloud users submit and the time taken to complete the task as well. By optimising resource allocation and utilisation, task scheduling is crucial to enhancing the effectiveness and performance of a cloud system. In this context, cloud computing offers a wide range of advantages, such as cost savings, security, flexibility, mobility, quality control, disaster recovery, automatic software upgrades, and sustainability. According to a recent research survey, more and more tech-savvy companies and industry executives are recognize and utilize the advantages of the Cloud computing. Hence, as the number of users of the Cloud increases, so did the need to regulate the resource allocation as well. However, the scheduling of jobs in the cloud necessitates a smart and fast algorithm that can discover the resources that are accessible and schedule the jobs that are requested by different users. Consequently, for better resource allocation and job scheduling, a fast, efficient, tolerable job scheduling algorithm is required. Efficient Hybrid Job Scheduling Optimization (EHJSO) utilises Cuckoo Search Optimization and Grey Wolf Job Optimization (GWO). Due to some cuckoo species' obligate brood parasitism (laying eggs in other species' nests), the Cuckoo search optimization approach was developed. Grey wolf optimization (GWO) is a population-oriented AI system inspired by grey wolf social structure and hunting strategies. Make span, computation time, fitness, iteration-based performance, and success rate were utilised to compare previous studies. Experiments show that the recommended method is superior.

**Funding:** The author(s) received no specific funding for this work.

**Competing interests:** The authors have declared that no competing interests exist.

## 1. Introduction

Cloud computing has completely revolutionised business since it enables the efficient pooling of computing resources. Cloud users can provision and distribute pay-per-use cloud computing resources using the Cloud Service Provider's public interface [1]. Recent advancements in cloud computing enable numerous geographically distributed and interconnected cloud data centres to provide pay-per-use on-demand services to cloud customers more efficiently [2]. According to [28], cloud data centres will handle 94% of computing workload by 2021. Cloud computing's novel concept has provided various benefits, including decreased infrastructure costs, execution time, and maintenance expenses, among others. However, the increased strain imposed by the execution of several cloud-based applications led to a decline in resource utilisation and a reduced return on investment [3]. Incorrect job scheduling among virtual machines is one of the key causes of a decline in cloud computing resource utilisation, resulting in a loss of processing performance. Therefore, task scheduling is essential in cloud computing to assure optimal resource use by providing acceptable performance under varying task restrictions, including execution deadlines.

In cloud computing, a variety of tasks may need to be programmed on a large number of virtual machines [4] to save development time and enhance system performance. Therefore, work planning is essential for restoring the adaptability and dependability of cloud-based solutions. Scheduling tasks, on the other hand, has a broad scope of optimization and greatly contributes to the development of dependable and adaptable dynamic solutions. The majority of cloud computing work scheduling algorithms are rule-based because they are simple to build. Rule-based algorithms perform badly in the preparation of multidimensional jobs. Moreover, resource allocation and scheduling are not only associated with quality of service (QoS), but can also have a long-term effect on the revenue of cloud service providers. Researchers have access to a wide variety of alternatives for resource scheduling, and resource scheduling is currently recognised as one of the most important concerns in the field of cloud computing.

Job scheduling assigns user-supplied tasks to the correct cloud virtual machine [5]. Cloud consumers must sign a service level agreement with the cloud provider to stipulate service quality, execution timetable, budget, and work security. The user may request the computer resources needed to finish his job in compliance with the SLA [6]. The performance of cloud computing is directly affected by task scheduling. With proper work scheduling, more money can be generated, performance can be enhanced, and SLA violations may be minimised. Due to the rising complexity of cloud computing, the scheduling problem has become increasingly difficult to solve. In a cloud computing context, however, devising an effective plan to solve the issue of job scheduling becomes more difficult.

To solve the cloud computing job scheduling problem, enumeration methods or heuristic-based solutions [7] may be used. This type of problem can be seen in the cloud computing job scheduling quandary. In the context of cloud computing, enumeration processes are not relevant since they require the generation of every possible combination of work schedules before selecting the most effective one. This is not possible with cloud computing. This method is laborious, which renders it inappropriate for use in a cloud computing setting with a significant amount of work to be done.

This study's primary objective is to search for and locate all accessible cloud resources, then rapidly distribute them in accordance with cloud users' job requirements. In this instance, we begin by randomly assigning the solution based on the number of jobs and cloud nodes. Each solution's goal function is then determined. This work develops a time- and quality-based objective function. After calculating fitness, we modify the solution using optimization techniques. Following the completion of the Cuckoo Search Optimization (CSO) process, the

EHJSO technique then uses the Grey Wolf Job Optimization (GWO) algorithm to allocate the resources that have been found to be available. The following is a list of the significant contributions that this work has made.

- The efficient Hybrid Job Scheduling Optimization (EHJSO) is developed to facilitate the cloud job scheduling process. In this, we create a multiobjective fitness function to help us with the scheduling method.

- The Cuckoo Search Optimization (CSO) and Grey Wolf Job Optimization (GWO) methods are hybridised and used to find the best solution, alleviating some of the issues inherent in the individual Cuckoo Search Optimization (CSO) and Grey Wolf Job Optimization (GWO) techniques.

- This method reduces total job completion time while maintaining quality standards and adhering to schedule constraints.

This study says integrating CSO and GWO helps with scheduling (EHJSO). This study continues as follows. Section 2 discusses important studies. Third section describes recommended approach. 4 discusses simulation results. Part 5 concludes.

## 2. Literature survey

Sobhanayak Srichandan et al. [8] suggested a hybrid strategy for cloud job scheduling that employs generally available biologically inspired heuristic algorithms. To enhance manufacturing time while reducing energy use, a scheduling technique was devised. The computing cost of the scheduling method, on the other hand, was extraordinarily expensive.

Ratin Gautam et al. [9] established a work scheduling GA. The GA task scheduler identifies the GA scheduling function for each cycle. Implement job assignments and evaluate task schedules using user satisfaction and virtual machine accessibility. The function iterated genetic functions to determine the best work plan. The operation was scheduled to minimise execution time and delay cost to lower total processing costs. Optimization ignored multiobjective functions. T. Prem Jacob and K. Pradeep developed Cuckoo Search and Particle Swarm Optimization to reduce missed deadlines. However, the remaining QoS settings weren't optimised.

The Dynamic Resource Allocation System for Cloud Computing is discussed in Saraswathiet et al. [10]. 's research paper. These studies highlight the distribution of virtual machines (VM) to users based on an analysis of workload factors. The basic concept of this work is that lower priority jobs (with a big task limit) should not impede the implementation of higher priority works (with a short deadline), and that VM resources for a consumer job should be assigned with zeal to achieve the deadline. In addition, because meeting these criteria is challenging, scheduling solutions do not take into account the dependability and accessibility of the cloud computing environment.

A hybrid cloud resource provisioning technique that blends autonomic computing with aa was proposed by Mostafa Ghobaei-Arani et al. A hybrid cloud resource provisioning technique that combines autonomic computing and reinforcement learning (RL) was proposed by Mostafa Ghobaei-Arani and colleagues [11]. The cloud layer concept created an autonomic resource provisioning architecture to help with the control Monitor, Analysis, Plan, and Execute (MAPE) cycle.

Hadeel Alazzam et al. [12] proposed a hybrid Tabu-Harmony technique for cloud computing job scheduling. Tabu and Harmony searches were created to improve the quality of search results. The throughput was raised while the makespan and total cost were reduced. Despite this, it was unable to increase process efficiency through improved design. The bin packing dilemma can be used to illustrate the scheduling dilemma.

A approach for load balancing that combines firefly and improved multiobjective particle swarm optimization was proposed by Francis Saviour Devaraj and colleagues [13]. [Citation needed] (IMPSO). The search space was narrowed down because to the development of Firefly (FF). After some time had passed, the IMPSO approach was developed in order to determine the improved response. It was determined through the use of the IMPSO algorithm that the ideal particle should have the shortest distance from point to line (gbest). The distance between a point and a line was minimised to get the best particle possibilities. To maximise resource consumption and reaction time, a suitable average load was achieved. However, memory and cost were not addressed in order to increase scheduling performance.

The execution time and cost were optimised by Zong-Gan Chen et al. [14] using cloud-based process scheduling. The architecture of a multi-purpose ant colony system was designed using a number of convolutional populations. The optimization target is established using non-dominated solutions from a global library, and a fresh pheromone updating strategy is created. A complementary heuristic technique for colony restriction and pheromone update rule support was created to keep the search in balance. On the other hand, performance evaluation in multi-cloud systems was not successfully adapted by cloud environments.

G. N. Reddyet et al. [15] introduced the whale optimization approach as a solution to the cloud data centre work scheduling problem. They evaluated resource utilisation, energy use, and service quality as fitness indicators. The whale optimization technique is used to optimise the suggested fitness function. The simulation findings indicate that the whale optimization method outperforms alternative methods for reducing energy consumption and optimising resource use while maintaining the service level agreement's needed service quality.

Mahendra Bhatu Gawali and his fellow workers came up with a heuristic methodology [16]. [Citation needed] It utilised the Modified Analytic Hierarchy Process (MAHP) with Bandwidth Aware Divisible Scheduling for the purpose of work scheduling and resource allocation (BATS). Methodologies such as divide-and-conquer, divide-and-conquer optimization, and the longest anticipated processing time pre-emption (LEPT) are also utilised. Before being allocated to cloud resources, each work was regulated using an MAHP technique. The BATS + BAR optimization method was then used to disperse the resources [17]. There was also discussion of the bandwidth and load limitations of cloud services. By leveraging LEPT preemption, resource-intensive tasks were also avoided. However, neither the scheduling nor the load distribution were optimised by the method. An efficient work scheduling mechanism is needed to manage a pool of virtual machines in a cloud computing data centre while ensuring service quality and resource efficiency. Undoubtedly, VM failure lowers the overall system throughput. A VM recovery technique that enables VMs to be copied to another host [18] can be used to solve this issue. According to Jin et al. [19], the scheduling of virtual machines (VM) is a crucial component for the efficient administration of computing resources in a data centre. When compared to alternative resource provisioning, resource utilisation rose [20]. However, the multi-tiered cloud systems described in [21] did not include admission control approaches. according to Y. The idea of cloud virtualization technology, according to Ling et al. [22], enables users to access computer resources through virtual machines (VM) and rent them to businesses or individual users. As cloud issues decrease performance, they should be controlled using the fault-tolerance technique [23]. similar to K. and S. Gupta According to Deep [24], the problem might have been brought on by bad hardware, a broken virtual machine, network congestion, or a broken application. A Hybrid GGWO algorithm optimization has been proposed [25] which combines the crossover and mutation operations of GA with the exploration and exploitation stages of GWO. This work proves the superiority of GGWO with high accuracy compared with different optimization algorithms in terms of Root Mean Squared Error (RMSE) and computational time.

Due to the discontinuous nature of the solution to the typical FMS scheduling problem, Tawhid et al. technique.'s has been slightly modified in its Levy Flight operator to select the optimal task. It is feasible to believe that the principal advantages of cloud computing—quantifiable services, adaptability, and resource polling—offer the best solution to problems with cost, compatibility, and a lack of IT personnel [26].

As a result of the economies of scale that are made feasible by cloud computing, a rising number of enterprises are deciding to install their applications in data centres that are hosted in the cloud [27]. Scheduling all incoming jobs while adhering to the service delay bound is difficult for a private cloud [28]. Visuwasam and L. Maria Michael [29] developed a method for the scheduling of work that maximises the profit of a private cloud while retaining the associated latency limitation. This scheduling issue was created and resolved using a heuristic approach. Based on application requirements, Devi. K et al. [30] proposed a way to optimise data centre resources and enable cloud computing.

A. Sermakani [31] put out an idea for a method of load balancing and scheduling that ignored the sizes of individual jobs. The author took into account the server's refresh times when processing queries. Balachander, K. [32] introduced task scheduling that takes bandwidth into account as a resource. The development of a nonlinear programming paradigm has made resource allocation between tasks possible. Priority-based job scheduling for cloud computing was proposed by Praveen D. S. [33]. When making judgments, a multiplicity of characteristics and features are evaluated. The primary focus of Subramani, Neelakandan, and their colleagues' work [34] is task scheduling that takes numerous restrictions into consideration. Due to the current state of the art, the authors of this study are inspired to continue their investigation into the scheduling of tasks and the allocation of resources.

## 3. Proposed system

### 3.1. Job scheduling in cloud computing

There are numerous sites from which you can access the cloud resources. They won't be put in the same location. There may be numerous tasks associated with the user-supplied query, and each job may require its own set of resources to run. The task scheduler assigns jobs to the appropriate resources. Fig 1 depicts the cloud-based job scheduling architecture.

In Fig 1, "U1, U2,. . . UN" represents the users, "W1, W2,. . . WN" represents the works or queries supplied by the users, "J1, J2,. . . JM" represents the tasks required to complete each work or query, and "R1, R2,. . . RP" represents the resources required to complete each work or query. Consider the following illustration: During the completion of a task, a user conducts a search that may include many occupations. Each task will be accomplished with a unique set of resources. The resources could be stored anywhere in the cloud. The job scheduler must assign the task to the right resource. An effective technique is required to correctly schedule jobs in a cloud computing environment. For scheduling cloud-based workloads, a novel resource searching and allocation method is proposed and is depicted in Fig 2. The Efficient Hybrid Job Scheduling Optimization (EHJSO) algorithm generates preliminary solutions based on user-supplied queries. The Cuckoo Search Optimization (CSO) and Grey Wolf Job Optimization (GWO) algorithms are then used to evaluate each solution, and the EHJSO algorithm obtains the best solutions from the CSO and GWO algorithms. The final optimal solution is utilised to schedule jobs after EHJSO performs search and find the optimal resource which are suitable to execute the given task.

In the part that follows, we will discuss the hybrid strategy that was developed in order to successfully plan jobs in an environment based on cloud computing. Fig 2 depicts the many actions that should be taken in order to implement the proposed plan.

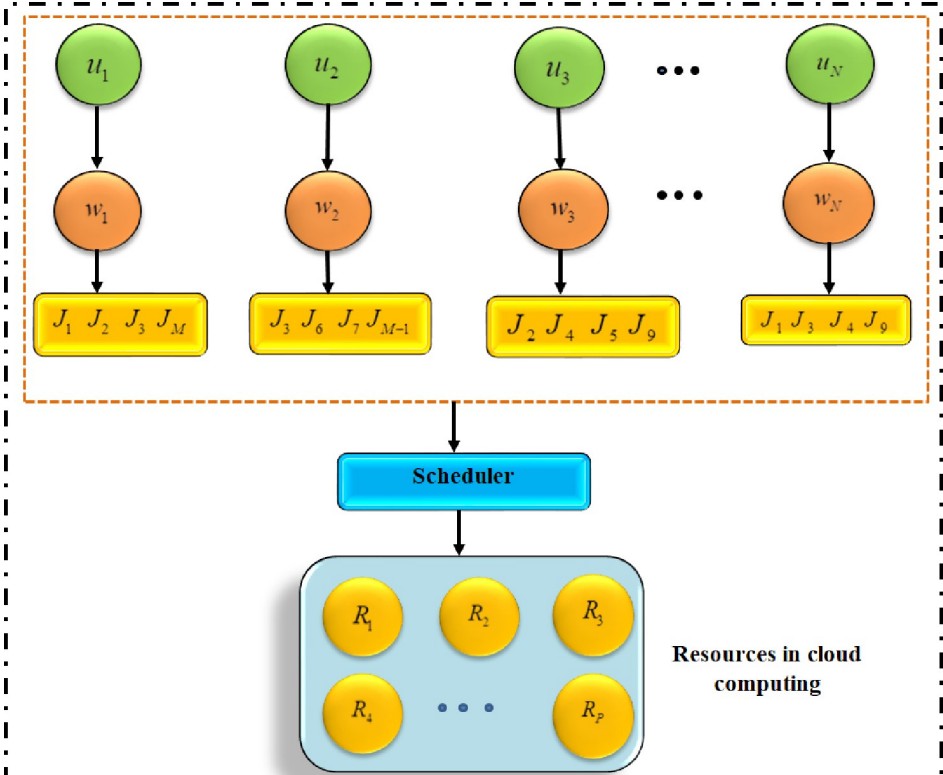

**Fig 1. Architecture of job scheduling in cloud.**

As shown in Fig 2, the user-supplied queries are initially used to develop preliminary solutions for the hybrid EHJSO algorithm. The best EHJSO solutions are then sent to the algorithms Cuckoo Search Optimization (CSO) and Grey Wolf Job Optimization (GWO). The jobs are scheduled according to the optimal solution determined by the Cuckoo Search Optimization (CSO) and Grey Wolf Job Optimization (GWO) algorithms.

**3.1.1. Preliminary solutions.** Based on the user's inquiries, the initial solutions for processing the hybrid EHJSO algorithm are generated. The user submits a query to finish a task, and the task is completed in accordance with the user's specifications. Numerous jobs will be executed by the query, each of which will use a different set of resources. It should be recognised that not every resource can complete every task at once. Although different resources can complete the same task, their efficiency and effectiveness will differ. The execution time and resource quality for a sample of jobs are shown in Table 1.

The following is explained in Table 1: Both resources R1 and R3 can be used to fulfil Job J1, however R1 requires nine seconds and has a quality of 4, while R3 requires five seconds and has a quality of 2. The resource R2 can be used to finish the task J2 in seven seconds with a quality rating of 2. The remaining tasks will also be finished with the proper tools. Table 1 is used to build preliminary solutions based on the user-provided jobs' ability to execute. Take a look at the following projects or questions that three users, U1, U2, and UN, submitted:

$$W_1 = \{J_1, J_2, J_4\} \tag{1}$$

$$W_2 = \{J_1, J_2, J_3\} \tag{2}$$

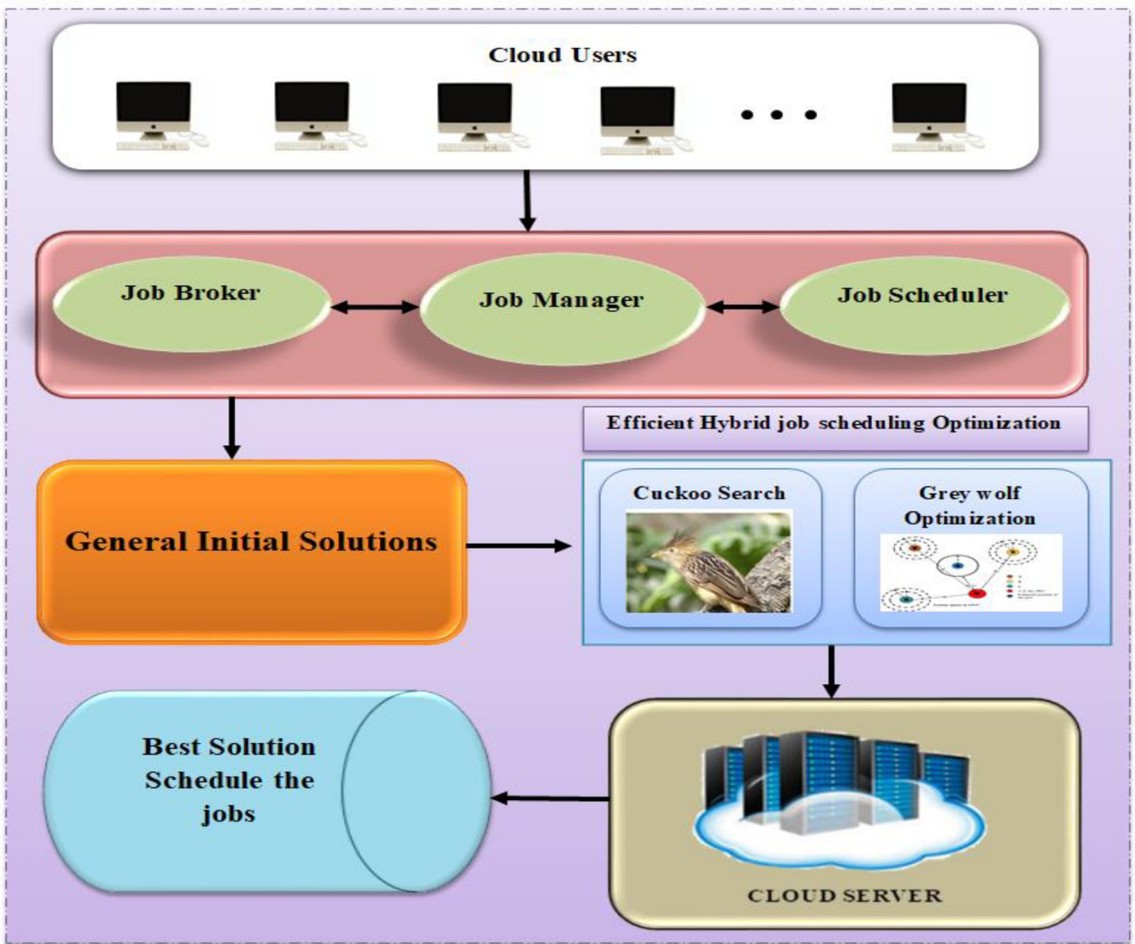

**Fig 2. Proposed diagram of EHJSO method.**

$$W_N = \{J_1, J_3, J_4, J_M\} \tag{3}$$

In the equation that was just presented, in order to fulfil the user's first query, the jobs $\{J_1, J_2$ and $J_4\}$ need to be carried out; in order to fulfil the user's second query, the jobs $\{J_1, J_2$ and $J_3\}$ need to be carried out; and in order to carry out the user's Nth query, the jobs $\{J_1, J_3, J_4$ and $J_M\}$ need to be carried out. a response that was produced by making use of these user-provided queries.

**Table 1. Sample time and resource quality to execute the job.**

|  | $R_1$ | $R_2$ | $R_3$ | - | $R_P$ |
|---|---|---|---|---|---|
| $J_1$ | (9,4) | - | (5,2) | - | - |
| $J_2$ | - | (7,2) | - | - | - |
| $J_3$ | - | - | - | - | - |
| $J_4$ | (4,2) | - | - | - | - |
|  | - | - | - | - | - |
| $J_M$ | (3,2) | - | - | - | - |

**3.1.2 Fitness evaluation.** Each produced solution is evaluated based on the resource's suitability for the given task. Calculating one's level of physical fitness takes into account a variety of factors, including the total amount of activities to be carried out., the amount of time needed to complete the tasks, and the quality of the resources used to complete the jobs. Using the following Eq (4), fitness is determined:

$$fit(S_i) = \alpha\left(\frac{T_{max}}{T_{total}}\right) + \beta\left(1 - \frac{\sum P}{M \times P_{max}}\right) \tag{4}$$

In the above equation, fit ($S_i$) represents the fitness of the i[th] solution; $T_{max}$ is the maximum amount of time a resource can spend to complete a task; and $T_{total}$ is the total amount of time required by resources to accomplish all tasks. P represents resource quality, while M represents the total number of tasks. Two tasks from different users cannot be executed concurrently on the same resource, nor can two jobs from the same user be executed concurrently within the same time frame.

## 3.2 Cuckoo search algorithm

A population-based metaheuristic stochastic global search approach is known as the cuckoo search (CS). In a computer science algorithm, cuckoo eggs stand in for possible solutions. The CS algorithm is exemplified by three rules, which are as follows: One egg at a time is laid by each cuckoo in a nest that has been chosen at random. The next generation will inherit the most comfortable nest with eggs of the highest possible quality (solutions). The number of host nests that are currently available is restricted, and there is a probability $P_a \in [0,1]$ that each host will discover at least one alien egg. If a host finds a cuckoo egg, the egg might be discarded, or the host might donate the egg to an invading species instead of keeping it in its own nest.

$P_a$ of $n$ $x^{(t+1)}$ for

To simplify matters, the fraction of $P_a$ of $n$ nests replaced with new nests storing random solutions can approximate the final hypothesis. The fitness of a maximisation solution and its objective function may be inversely connected. Like other evolutionary algorithms, multiple fitness types can be established. Fresh solutions are represented by cuckoo eggs. The goal is to replace substandard nest solutions with new ones (cuckoos). In Eq (5), a Levy flight is used to generate new solutions for $x^{(t+1)}$ cuckoo i.

$$x_i^{(t+1)} = x_i^t + \alpha \oplus Le\prime vy(\lambda) \tag{5}$$

where $\alpha(\alpha>0)$ represents a step scaling size. This parameter ought to have some sort of connection to the magnitude of the issue that the algorithm is attempting to resolve. In the vast majority of situations, it is possible to set to the value 1 or another constant. The term "product" $\oplus$ refers to the process of multiplying entries entry-by-entry. The duration of the random step is chosen at random from a distribution that has an infinite mean but an infinitely variable standard deviation.

$$Le\prime vy\ u = t^{-\lambda} \tag{6}$$

where $0 \leq \lambda \leq 3$. In this case, the cuckoo's succession of hops or steps forms the essence of a random walk process, one that complies with a power-law step-length distribution with a heavy tail.

```
Algorithm 1:Cuckoo search Algorithm
Initialization
Objective function f(x⃗), x⃗ = (x₁,x₂,...,x_d)ᵀ
Generation t = 1
Initial population of n host nests x_i(i = 1,2,..,n)
```

```
While (t<(Maximum Generation) or (stop criterion))
    Get a cuckoo (say i) randomly by Levy flights
    Evaluate fitness for cuckoo F
    Choose a nest among n (say j) randomly
    If (Fᵢ>Fⱼ) then
        Replace j by the new solution
    End if
    Abandon a function (Pₐ) of worse nests and build new ones
    Keep the best solutions (or nests with quality solution)
    Rank the solution and find the current best
    Update the generation number t = t+1
End While
```

## 3.3. Grey Wolf Optimisation algorithm

The University of California at Berkeley is responsible for the development of the swarm-based algorithm known as GWO. Its social behaviour in the wild is fashioned after that of grey wolves in their natural environment, which is a good thing. The pursuit of prey by wolves, through stalking and hunting, demonstrates the pursuit of the most advantageous option. While living in the wild, grey wolves prefer to congregate in groups known as packs. Wolf packs are typically comprised of five to twelve individuals in size. Additional to this, the wolves in the pack are divided into four groups according to their social position, which makes the hunting process easier. The following are the names of the groups: In a dog pack, the Alpha ($\alpha$), who can be either male or female, is the pack's leader and is in charge of making decisions about hunting, waking up, and sleeping. Beta ($\beta$) is the second level of wolves, and it is made up of either male or female wolves who assist the other wolves in the pack with decision-making and group decision-making. Delta ($\delta$), the third rank, is responsible for a variety of important responsibilities such as carer, sentinel, pack leader, and hunter. Omega ($\omega$) is the final and most difficult level to reach. Despite being the weakest of the levels in the hierarchical paradigm, this one serves as a scapegoat who must obey by the mandates of the higher-ups.

**3.3.1. The mathematical model of Grey Wolf Optimization (GWO) algorithm.** Grey wolves have a social order, and their hunting strategy is like that of grey wolves. There are four levels in it.

*Level-1*: Alpha (α): It controls how decisions are made (for example, decisions about shooting, wake-up time, and slumber location).

*Level-2*: Beta (β): The contender has the best chance of succeeding the wolf as the new leader. acts as an advisor or consultant for

*Level-3*: Delta (δ): Which wolves have earned their respect at this time are identified to the wolves. They are searching for x wolves. They perform a variety of roles, including scouts, sentinels, elders, group carers, and predators.

*Level-4*: Omega (ω): They believed that the thinnest wolves were 1/4 equal wolves. They assume the role of accuser.

**3.3.2. Mathematical model and algorithm.** Each of the three components of the GWO which relate to the mathematical model is described below. Encirclement, hunting, and attachment are all examples of behaviours that occur in this situation. Surrounding behaviour is signified by the calculation under encircling behaviour as the formulas shown below.

$$\overrightarrow{D} = |C. \overrightarrow{X_P}(t) - \overrightarrow{X}(t)| \tag{7}$$

$$\overrightarrow{X}(t+1) = \bar{X}(t) - \overrightarrow{A}.\overrightarrow{D} \tag{8}$$

where t represents the already present repetition, (A and C) stand for two constant matrix vectors, Xp. represents the location vector of the prey, and X stands for the location vector of the grey wolf. The following provides a description of the vectors (A and C):

$$\overrightarrow{A} = 2\overrightarrow{d}.r_1 - \overrightarrow{d} \tag{9}$$

$$\overrightarrow{C} = 2\overrightarrow{d}.\overrightarrow{r_2} \tag{10}$$

Over iterations, a fall in values from 2 to 0 may occur *(further it is explained in Eq (18))*, and r1, r2 are random vectors. Second, he grey wolves can locate quarry and quest it, and they often shadow the important wolf through this procedure. In adding, the beta edition and estuary may join in shooting on a limited basis. Even while beta and delta wolves are very familiar with the locations of potential food, the alpha wolf is usually seen to be the best choice. In order to update the locations of the extra wolves, including the completion wolf, the locations of the important, beta, and delta wolves will therefore be used, as illustrated in the calculations below.

$$\overrightarrow{x}(t+1) = \frac{1}{3}\overrightarrow{X_1} + \frac{1}{3}\overrightarrow{X_2} + \frac{1}{3}\overrightarrow{X_3} \tag{11}$$

where $X_1$, $X_2$ and $X_3$ are given by following Equations:

$$\overrightarrow{X_1} = \overrightarrow{X_\alpha}(t) - \bar{A_1}.\overrightarrow{D_\alpha} \tag{12}$$

$$\overrightarrow{X_2} = \overrightarrow{X_\beta}(t) - \bar{A_2}.\overrightarrow{D_\beta} \tag{13}$$

$$\overrightarrow{X_3} = \overrightarrow{X_\delta}(t) - \bar{A_3}.\overrightarrow{D_\delta} \tag{14}$$

$X_\alpha$, $X_\beta$ and $X_\delta$ are the alpha, beta, and delta wolves' iteration positions, or the first three effective answers to the problem. They are A1, A2, and A3.

$D_\alpha$, $D_\beta$ and $D_\delta$ are given by the following Equations:

$$\bar{D}_\alpha = |\overrightarrow{C_1}.\overrightarrow{X_\alpha} - \overrightarrow{X}| \tag{15}$$

$$\bar{D}_\beta = |\overrightarrow{C_2}.\overrightarrow{X_\beta} - \overrightarrow{X}| \tag{16}$$

$$\bar{D}_\delta = |\overrightarrow{C_3}.\overrightarrow{X_\delta} - \overrightarrow{X}| \tag{17}$$

where $C_1$, $C_2$, and $C_3$ are represented by the Eqs (7) and (8), respectively. It demonstrates how the position of the search wolf in the search space moves when the values of alpha, beta, and delta change over time. Particularly noteworthy is the fact that the final site would be an accidental point within a round shaped by the search space's alpha, beta, and delta values. Simply said, the alpha, beta, and delta wolves decide the location of the quarry, while the other wolves inform their locations in the vicinity of the kill at random intervals.

When the prey has come to a complete stop, the wolves will launch their attack. During an iteration, a fall in value from 2 to 0 indicates that the wolves are close to their prey and should be avoided. The following equation expresses the value of the fraction:

$$\overrightarrow{a} = 2 - \frac{2*t}{Maxltr} \tag{18}$$

where t is the existing recurrence and *MaxItr* characterizes the supreme number of repetitions. In this study, the GWO procedure was chosen as a feature assortment instrument. Because of its behaviour based on meta-heuristics, it can find the best answer and avoid stacking on one option. It performs admirably in hitherto unexplored and difficult search locations. It also has a small number of control settings and is simple to implement. Furthermore, the GWO bases its selection on the top three search agents. In the initialisation step, the subsequent Comparition is used to transform the explanation created by the GWO procedure to binary standards, and in the final stage, the subsequent Comparation is used to select the best explanation as shown below.

$$Z_i = round(|y_i \bmod 2|) \bmod 2 \tag{19}$$

$Z_i$ denotes the binary worth (discrete value) characterized by 0 or 1 and $y_i$ denotes the rate of the explanation (continues values) produced by the initialization and concluding phases of the algorithm. According to Eq (19), the binary number is zero if the complete rate of the residual falls between 0 and 0.4999, or between 1.5 and 1.9999, whereas the binary number is 1 otherwise. For a binary number that is between 0.5 and 1.4998, the complete rate of the remaining falls between these two ranges.

The first three optimal locations for the alpha, beta, and delta wolves will also draw the attention of the remaining wolves in the packet after one iteration of the algorithm, as seen in Fig 3. Alphas is the response (position) with the most correct categorization, followed by betas and deltas. At the end of repetition of the process, the classification algorithm is qualified and authenticated, and the accurateness of the classifier is intended for each subsection (solution) of the situation medium.

$$Fitness\ Function = W_1 * Accuracy + W_2 * \frac{1}{Number\ of\ Selected\ Resources} \tag{20}$$

A list of resources is included in each resource subset. If two subsets have the same accuracy but differ in the sum of characteristics, the subsection with the less structures will be chosen. Furthermore, the standards of $(W_1)$ and $(W_2)$ in an above calculation are configurable, with the disorder that $(W_1)$ be increased by Correctness and $(W_2)$ by the opposite Sum of Designated Structures.

## 4. Result and discussion

### 4.1. Experimental setup

In this article, the usefulness of the suggested job scheduling technique is reviewed and analysed. To test the suggested technique for scheduling work using Java (jdk 1.8) and CloudSim, a workstation with an i5 processor running at 2.30 GHz, 8 GB of RAM, and a 64-bit version of Windows 10 was utilised.

### 4.2 Performance analysis and comparison

By varying the number of iterations and the preliminary solution provided in response to three distinct inputs, the performance of the proposed EHJSO technique is compared to that of alternative job scheduling algorithms in terms of fitness achieved and time needed to execute the scheduled jobs.

### 4.3. Makespan

Table 2 and Fig 4 present in full the comparison analysis that Makespan performed between the EHJSO methodology and other methodologies. The result shows that the EHJSO method has outperformed the other techniques in all aspects. For example, with 20 number of Iterations, the EHJSO method has taken only 96.14 sec to respond, while the other existing

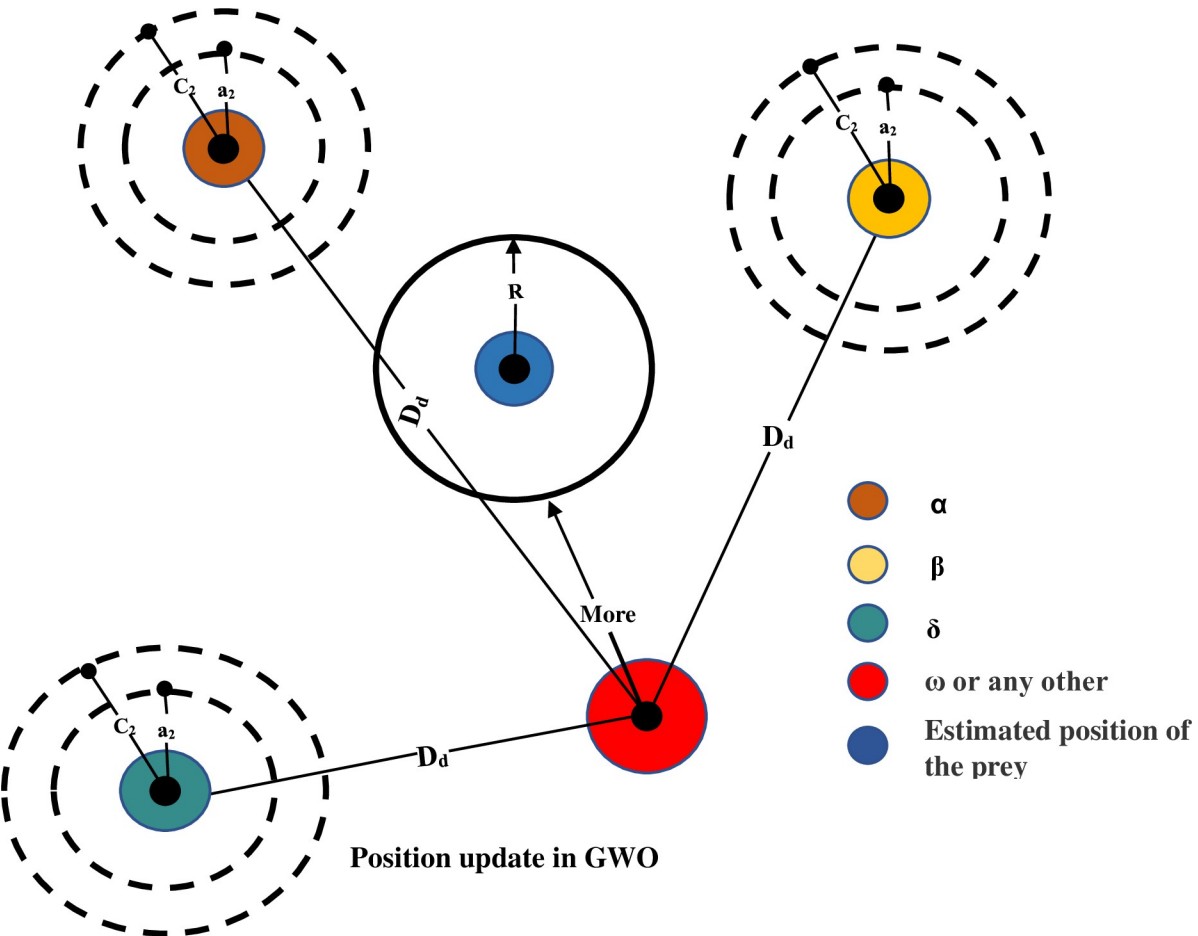

**Fig 3. GWO iterative updates in position.**

techniques like BAT, WBAT, Firefly and BLA have a Makespan of 110.37sec, 106.38sec, 102.35sec and 98.15sec respectively. Similarly, for 100 number of Iterations, the EHJSO method has a Makespan of 95.83 sec while the other existing techniques like BAT, WBAT, Firefly and BLA have 118.54sec, 108.37sec, 103.86sec and 100.98sec of Makespan, respectively.

## 4.4. Computation time

In comparison to other methodologies, the calculation time analysis of the EHJSO methodology is displayed in Table 3 and Fig 5. The data clearly shows that the EHJSO method has outperformed the other techniques in all aspects. For example, with 20 number of Iterations, the

**Table 2. Makespan analysis for EHJSO method with existing system.**

| No of Iterations | BAT algorithm | WBAT Job Scheduler | Firefly Algorithm | Bees life algorithm (BLA) | EHJSO |
|---|---|---|---|---|---|
| 20 | 110.37 | 106.38 | 102.35 | 98.15 | 96.14 |
| 40 | 113.75 | 107.28 | 101.76 | 99.34 | 97.37 |
| 60 | 112.87. | 106.95 | 102.86 | 98.87 | 96.84 |
| 80 | 114.64 | 107.84 | 103.43 | 100.46 | 95.27 |
| 100 | 118.54 | 108.37 | 103.86 | 100.98 | 95.83 |

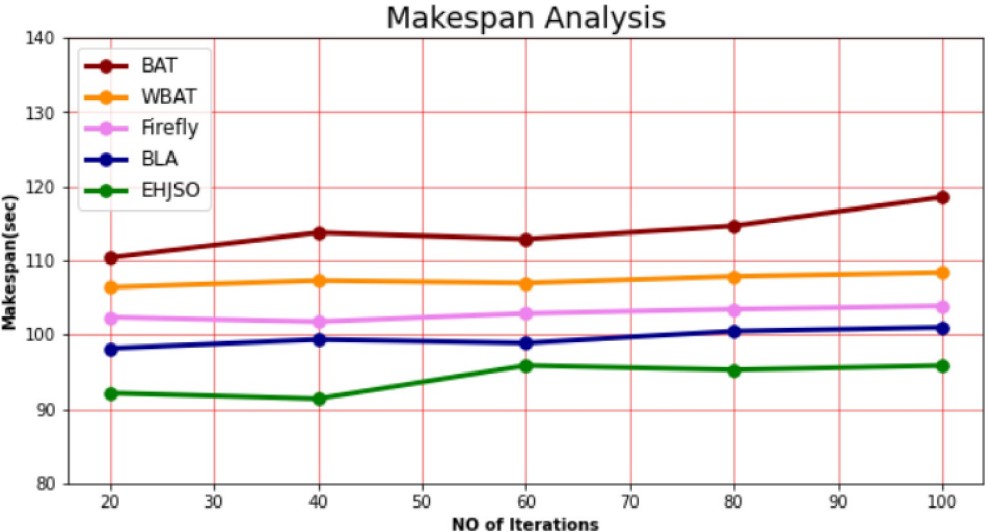

**Fig 4. Makespan analysis for EHJSO method with existing system.**

EHJSO method has taken only 216.65ms to respond, while the other existing techniques like BAT, WBAT, Firefly and BLA have a Computation time of 417.76ms, 336.75ms, 302.54ms and 263.76ms respectively. Similarly, for 100 number of Iterations, the EHJSO method has a Computation time of 251.54ms while the other existing techniques like BAT, WBAT, Firefly and BLA have 458.17ms, 392.87ms, 3 of Computation time59.27ms and 289.43ms, respectively.

### 4.5. Fitness

The fitness of the EHJSO technique is compared to other methods in Table 4 and Fig 6. The data clearly shows that the EHJSO method has outperformed the other techniques in all aspects. For example, with 20 number of Iterations, the EHJSO method has taken only 7.268 sec to respond, while the other existing techniques like BAT, WBAT, Firefly and BLA have a Fitness of 13.764sec, 11.437sec, 10.873sec, and 8.517sec respectively. Similarly, for 100 number of Iterations, the EHJSO method has a Fitness of 8.248 sec while the other existing techniques like BAT, WBAT, Firefly and BLA have 13.378sec, 13.427sec, 11.864sec and 10.817sec of Fitness, respectively.

### 4.6. Performance based on iterations

Table 5 and Fig 7 describe the analysis of Performance based on Iterations of the EHJSO technique with the existing methods. The findings make it abundantly evident that the suggested approach is superior to the various other methods in every respect. For example, with 20 Iterations, the EHJSO method has a Performance based on Iterations of 88.22% while the other

**Table 3. Computation time analysis for EHJSO method with existing system.**

| No of Iterations | BAT algorithm | WBAT Job Scheduler | Firefly Algorithm | Bees life algorithm (BLA) | EHJSO |
|---|---|---|---|---|---|
| 20 | 417.76 | 336.75 | 302.54 | 263.76 | 216.65 |
| 40 | 409.43 | 361.65 | 316.78 | 271.28 | 228.54 |
| 60 | 437.65 | 382.76 | 338.65 | 284.76 | 231.75 |
| 80 | 423.58 | 374.54 | 329.28 | 293.81 | 246.73 |
| 100 | 458.17 | 392.87 | 359.27 | 289.43 | 251.54 |

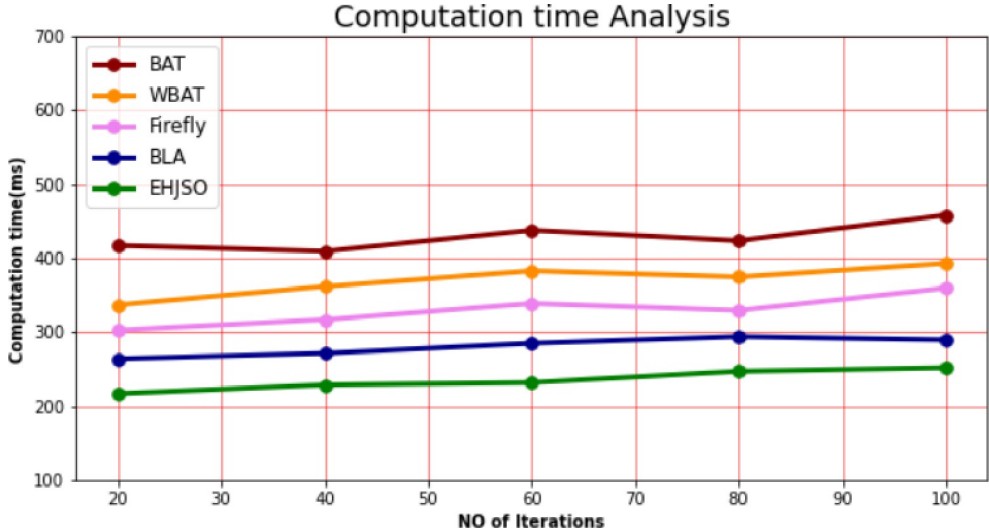

**Fig 5. Computation time analysis for EHJSO method with existing system.**

existing methods like BAT, WBAT, Firefly and BLA have a Performance based on Iterations of 76.92%, 77.95%, 80.87% and 83.69%, respectively. Similarly, with 100 Iterations, the proposed method has 93.86% of Performance based on Iterations while the other existing methods, BAT, WBAT, Firefly and BLA have a Performance based on Iterations of 78.74%, 81.54%, 83.28% and 86.28%, respectively. This proves that the EHJSO technique has higher performance with greater Performance based on Iterations.

## 4.7. Success rate

In Table 6 and Fig 8, the Success Rate of the EHJSO methodology in comparison to the current methods is shown. The data demonstrate that the suggested technique performs better than the alternatives in all respects. For example, with 20 Iterations, the EHJSO method has a Success Rate of 95.63% while the other existing methods like BAT, WBAT, Firefly and BLA have a Success Rate of 85.64%, 87.84%, 90.16% and 93.15%, respectively. Similarly, with 100 Iterations, the proposed method has 97.17% of Success Rate while the other existing methods, BAT, WBAT, Firefly and BLA have a Success Rate of 86.93%, 89.72%, 92.34% and 94.89%, respectively. This proves that the EHJSO technique has higher performance with greater Success Rate.

## 5. Conclusion

In this research, we propose a hybrid solution for task scheduling that combines the advantages of the Cuckoo search method and the Grey Wolf job optimization algorithm. Dominance

**Table 4. Fitness analysis for EHJSO method with existing system.**

| No of Iterations | BAT algorithm | WBAT Job Scheduler | Firefly Algorithm | Bees life algorithm (BLA) | EHJSO |
| --- | --- | --- | --- | --- | --- |
| 20 | 13.764 | 11.437 | 10.873 | 8.517 | 7.268 |
| 40 | 13.964 | 12.387 | 10.065 | 8.982 | 6.509 |
| 60 | 14.065 | 12.932 | 10.438 | 9.018 | 6.947 |
| 80 | 14.652 | 12.538 | 11.376 | 9.387 | 8.043 |
| 100 | 13.378 | 13.427 | 11.864 | 10.817 | 8.248 |

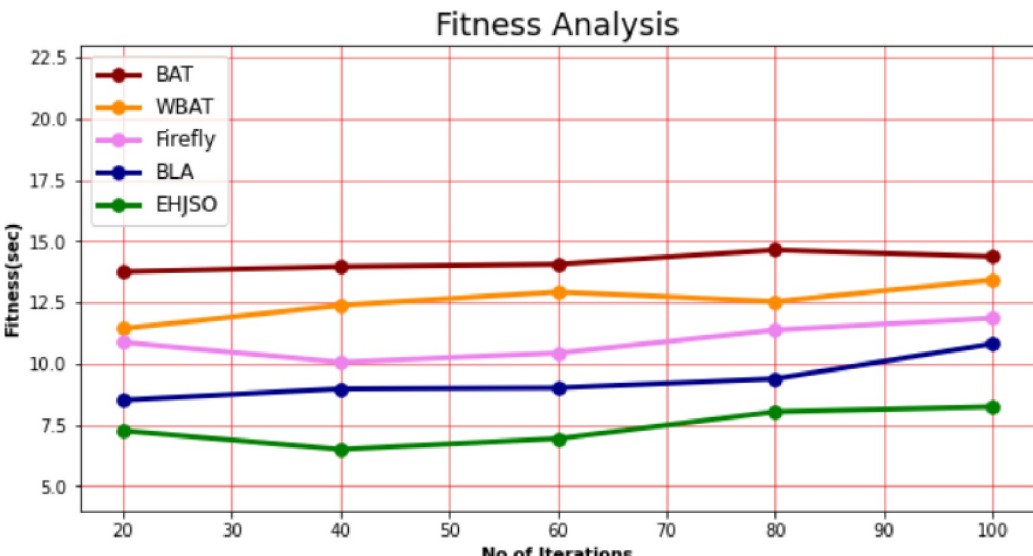

**Fig 6. Fitness analysis for EHJSO method with existing system.**

**Table 5. Performance based on Iterations analysis for EHJSO method with existing system.**

| No of Iterations | BAT algorithm | WBAT Job Scheduler | Firefly Algorithm | Bees life algorithm (BLA) | EHJSO |
|---|---|---|---|---|---|
| 20 | 76.92 | 77.95 | 80.87 | 83.69 | 88.22 |
| 40 | 77.13 | 79.16 | 82.31 | 84.32 | 80.86 |
| 60 | 77.53 | 79.52 | 81.84 | 85.87 | 91.65 |
| 80 | 78.28 | 80.42 | 82.74 | 84.93 | 90.54 |
| 100 | 78.74 | 81.54 | 83.28 | 86.28 | 93.86 |

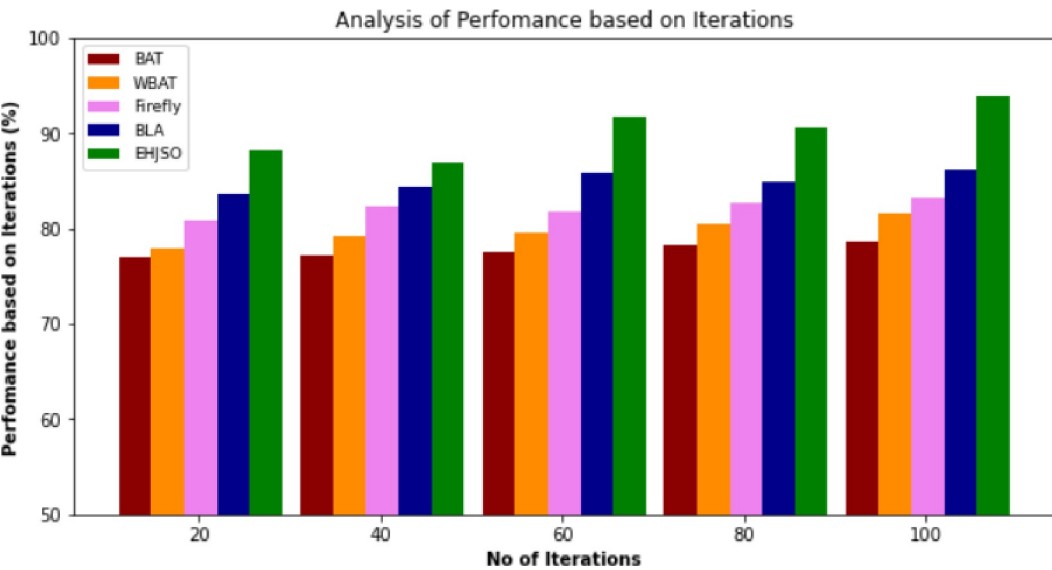

**Fig 7. Performance based on Iterations analysis for EHJSO method with existing system.**

**Table 6. Success rate analysis for EHJSO method with existing system.**

| No of Iterations | BAT algorithm | WBAT Job Scheduler | Firefly Algorithm | Bees life algorithm (BLA) | EHJSO |
|---|---|---|---|---|---|
| 20 | 85.64 | 87.84 | 90.16 | 93.15 | 95.63 |
| 40 | 86.27 | 88.13 | 91.21 | 92.85 | 95.93 |
| 60 | 85.73 | 88.62 | 90.84 | 93.85 | 96.13 |
| 80 | 87.18 | 89.19 | 91.85 | 94.32 | 98.67 |
| 100 | 86.93 | 89.72 | 92.34 | 94.89 | 97.17 |

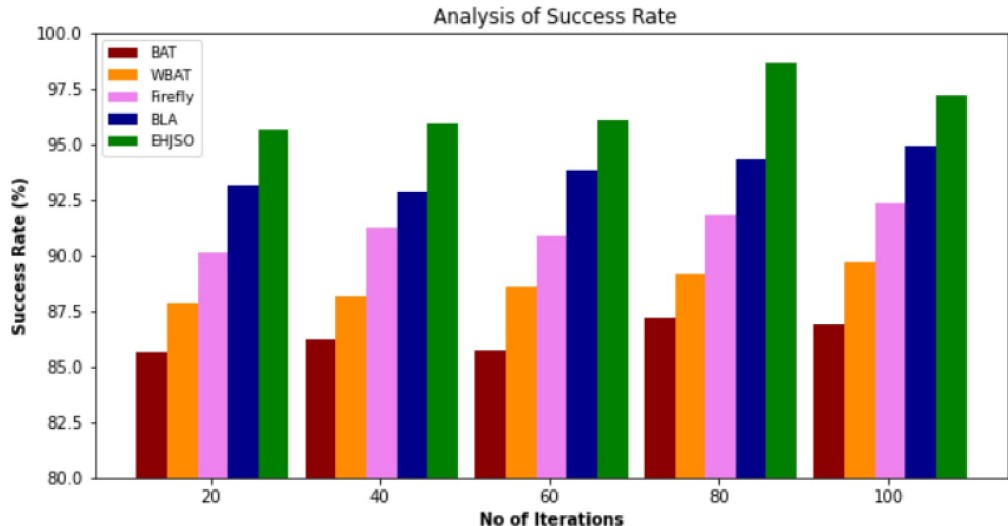

**Fig 8. Success rate analysis for EHJSO method with existing system.**

of these algorithm together yield best results as we have measured its performance using different metrics. The newly generated preliminary solutions are fed into the EHJSO algorithm so that it can carry out its intended function. The EHJSO algorithm would optimise job scheduling by taking into consideration execution time and the quality of currently available resources. Following that, the jobs that had been previously planned are carried out, and the user is presented with the output that is pertinent to the query that they submit. The proposed method EHJSO is evaluated and compared to other similar works of cloud resource allocation. The evaluation takes into account both the total amount of time needed to complete the given tasks and the level of fitness attained. The proposed method performed better than the competing algorithms in their performance, Both in terms of general health and the amount of time required to carry out the indicated duties, the equipment was insufficient. In order to emphasize the performance of EHJSO various relevant metrics are taken and compared with other works as well. More research is required to see whether the proposed method can be applied to a variety of large-scale and real-world optimization problems.

## Author Contributions

**Formal analysis:** E. Baburaj.

**Investigation:** S. Neelakandan.

**Methodology:** E. Baburaj.

**Resources:** T. Sethukarasi, M. Prakash.

**Software:** T. Sethukarasi.

**Writing – original draft:** D. Paulraj.

**Writing – review & editing:** D. Paulraj, S. Neelakandan, M. Prakash.

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
