## [Decision Letter · Decision Letter 0]

21 Dec 2022

PONE-D-22-30865An Efficient Hybrid Job Scheduling Optimization (EHJSO) Approach to Enhance Resource Search using Cuckoo and Grey Wolf job optimization for cloud environment.PLOS ONE

Dear Dr. Eppipanious,

Thank you for submitting your manuscript to PLOS ONE. After careful consideration, we feel that it has merit but does not fully meet PLOS ONE’s publication criteria as it currently stands. Therefore, we invite you to submit a revised version of the manuscript that addresses the points raised during the review process.

Two reviewers gave many useful comments. Please revise your manuscript according to their comments.

We look forward to receiving your revised manuscript.

Kind regards,

Shih-Wei Lin

Academic Editor

PLOS ONE

Journal Requirements:

5. Please amend the manuscript submission data (via Edit Submission) to include authors Dr.D.Paulraj and Dr.S.Neelakandan.

Additional Editor Comments:

Two reviewers gave many useful comments to improve your mansucript. Please revise your manuscript according to these comments.

Reviewers' comments:

Reviewer's Responses to Questions

**Comments to the Author**

1. Is the manuscript technically sound, and do the data support the conclusions?

Reviewer #1: Partly

Reviewer #2: Yes

2. Has the statistical analysis been performed appropriately and rigorously? 

Reviewer #1: Yes

Reviewer #2: Yes

3. Have the authors made all data underlying the findings in their manuscript fully available?

Reviewer #1: Yes

Reviewer #2: Yes

4. Is the manuscript presented in an intelligible fashion and written in standard English?

Reviewer #1: Yes

Reviewer #2: Yes

5. Review Comments to the Author

Reviewer #1: This manuscript combines the cuckoo search and grey wolf optimization for optimizing makespan, computation time, fitness, and success rate for job execution in cloud computing. Also, some simulation experiments are provided to verify the proposal's performance. It is interesting.

The Abstract is somewhat long, so I suggest compressing the introduction of the research background.

Job/workflow/task scheduling for cloud computing has attracted so much attention. The shortcomings of the existing works, such as https://ieeexplore.ieee.org/document/8443134, should be further analyzed.

As stated in the Abstract and Introduction, this manuscript does introduce a hybrid approach. But what are its new features? I suggest highlighting this aspect.

What is the allowable optimization delay when scheduling jobs in the public cloud? In some real-time application scenarios, the permissible delay is as low as milliseconds. But, the time overheads of cuckoo search and grey wolf techniques are considerable. It is necessary to investigate this respect.

Reviewer #2: In this paper, an efficient job scheduling algorithm, the Efficient Hybrid Job Scheduling Optimization, is proposed for better resource allocation and job scheduling. Then, Make span, computation time, adaptation, iteration-based performance and success rate are used to compare previous studies. The experimental results show that the method is superior. This work, appears to be applicable, but the paper seems to have some problems.

1.Please simplify the abstract section by suggesting that the author focus more on the innovative points of the paper and their contributions.

2.Please update the serial number of the citation, e.g. in the first paragraph of the introduction, that is, “According to [28], cloud data centres will handle 94% of computing workload by 2021. ”.

3.In the literature survey section, it is suggested that the author describe the relevant research results in recent years. I suggest the author to introduce some recent proposed meta-heuristics such as slime mould algorithm (SMA), Hunger Games Search (HGS), RUN Kutta optimizer (RUN), Colony Predation Algorithm (CPA) and Harris Hawk Optimization (HHO) to make the paper get more readers.

4.In the introduction section, the novelty of the work is missing. There is no sufficient justification as to why such a proposal is needed. It is recommended that the author elaborate on the shortcomings of the basic GWO and then elaborate on the main points about how your proposal will alleviate the existing problems of the GWO.

5.Please ask the author to standardize the abbreviations of the chart names. For example, it is suggested to change "Figure 1" to "Fig. 1".

6.Please ask the author to standardize the description of formula symbols in the manuscript. For example, formula letters should be uniformly italicized, etc.

7.The related work in section 2 is not clearly categorized. The related work section is too simplistic for this topic. To better highlight the related work, it is recommended that the author add a brief table of the main elements.

8.For the parameters α and β in Eq. (4), it is suggested that the author gives them in the text. Similarly, the parameters W1 and W2 in Eq. (20) are the same.

9.For Eq. (6), it is suggested that the author draws the levy distribution.

10.Please ask the author to recheck the sentence, that is, “The contender has the best chance of succeeding the wolf as the new leader. acts as an advisor or consultant for”.

11.In between the concluding chapters, it is advisable to have a separate discussion section to analyze the experiments conducted in this paper in depth, what are the advantages and disadvantages compared to the existing methods? What are the specific shortcomings? To facilitate a better tour for the reader.

12.Limitations of this work should be added in the conclusion section.

13.References should include DOI numbers, please reorganize and adjust them by the author.

6. PLOS authors have the option to publish the peer review history of their article (what does this mean?). If published, this will include your full peer review and any attached files.

Reviewer #1: No

Reviewer #2: No

---

## [Author Response · Author response to Decision Letter 0]

4 Feb 2023

We have incorporated all the changes recommended by the reviewers. We thank the anonymous reviewers for their valuable suggestions and help rendered to improve the clarity of the manuscript.

---

## [Decision Letter · Decision Letter 1]

21 Feb 2023

An Efficient Hybrid Job Scheduling Optimization (EHJSO) Approach to Enhance Resource Search using Cuckoo and Grey Wolf job optimization for cloud environment.

PONE-D-22-30865R1

Dear Dr. Eppipanious,

We’re pleased to inform you that your manuscript has been judged scientifically suitable for publication and will be formally accepted for publication once it meets all outstanding technical requirements.

Kind regards,

Shih-Wei Lin

Academic Editor

PLOS ONE

Additional Editor Comments (optional):

The quality of this manuscript has been improved significantly. Both reviewers suggest accepting the revised manuscript. Therefore, I recommend accepting the revised manuscript.

Reviewers' comments:

Reviewer's Responses to Questions

**Comments to the Author**

1. If the authors have adequately addressed your comments raised in a previous round of review and you feel that this manuscript is now acceptable for publication, you may indicate that here to bypass the “Comments to the Author” section, enter your conflict of interest statement in the “Confidential to Editor” section, and submit your "Accept" recommendation.

Reviewer #1: (No Response)

Reviewer #2: All comments have been addressed

2. Is the manuscript technically sound, and do the data support the conclusions?

Reviewer #1: Yes

Reviewer #2: Yes

3. Has the statistical analysis been performed appropriately and rigorously? 

Reviewer #1: Yes

Reviewer #2: Yes

4. Have the authors made all data underlying the findings in their manuscript fully available?

Reviewer #1: Yes

Reviewer #2: Yes

5. Is the manuscript presented in an intelligible fashion and written in standard English?

Reviewer #1: Yes

Reviewer #2: Yes

6. Review Comments to the Author

Reviewer #1: All my concerns have been well addressed. The revised version is very good, I recommend publishing it.

Reviewer #2: From the response letter, the paper has been well revised, and the current version of the manuscript is acceptable for publication.

7. PLOS authors have the option to publish the peer review history of their article (what does this mean?). If published, this will include your full peer review and any attached files.

Reviewer #1: No

Reviewer #2: No

---

## [Editor Report · Acceptance letter]

3 Mar 2023

PONE-D-22-30865R1 

An Efficient Hybrid Job Scheduling Optimization (EHJSO) Approach to Enhance Resource Search using Cuckoo and Grey Wolf job optimization for cloud environment. 

Dear Dr. Baburaj:

I'm pleased to inform you that your manuscript has been deemed suitable for publication in PLOS ONE. Congratulations! Your manuscript is now with our production department. 

Kind regards, 

on behalf of

Professor Shih-Wei Lin 

Academic Editor

PLOS ONE